# COVID-19 Diagnosis: A Comprehensive Review of the RT-qPCR Method for Detection of SARS-CoV-2

**DOI:** 10.3390/diagnostics12061503

**Published:** 2022-06-20

**Authors:** Debashis Dutta, Sarah Naiyer, Sabanaz Mansuri, Neeraj Soni, Vandana Singh, Khalid Hussain Bhat, Nishant Singh, Gunjan Arora, M. Shahid Mansuri

**Affiliations:** 1Department of Pharmacology and Experimental Neuroscience, College of Medicine, University of Nebraska Medical Center, Omaha, NE 68198, USA; 2Department of Microbiology and Immunology, University of Illinois at Chicago, Chicago, IL 60616, USA; snaiyer@uic.edu; 3IDEAS Dental College Gwalior, Gwalior 474020, MP, India; nazsaba82615@gmail.com; 4Department of Cell and Developmental Biology, University of Michigan, Ann Arbor, MI 48109, USA; sneeraj@med.umich.edu; 5Department of Pathology and Microbiology, College of Medicine, University of Nebraska Medical Center, Omaha, NE 68198, USA; vsingh@unmc.edu; 6SKUAST Kashmir, Division of Basic Science and Humanities, Faculty of Agriculture, Wadura Sopore 193201, JK, India; khalidbio@gmail.com; 7Cell and Gene Therapy Absorption System, Exton, PA 19335, USA; nishant900@gmail.com; 8Section of Infectious Diseases, Yale University School of Medicine, New Haven, CT 06520, USA; arorag1983@gmail.com; 9Molecular Biophysics and Biochemistry, Yale University, New Haven, CT 06511, USA

**Keywords:** SARS-CoV-2, COVID-19, pandemic, RT-qPCR, cDNA

## Abstract

The world is grappling with the coronavirus disease 2019 (COVID-19) pandemic, the causative agent of which is severe acute respiratory syndrome coronavirus 2 (SARS-CoV-2). COVID-19 symptoms are similar to the common cold, including fever, sore throat, cough, muscle and chest pain, brain fog, dyspnoea, anosmia, ageusia, and headache. The manifestation of the disease can vary from being asymptomatic to severe life-threatening conditions warranting hospitalization and ventilation support. Furthermore, the emergence of mutecated variants of concern (VOCs) is paramount to the devastating effect of the pandemic. This highly contagious virus and its emergent variants challenge the available advanced viral diagnostic methods for high-accuracy testing with faster result yields. This review is to shed light on the natural history, pathology, molecular biology, and efficient diagnostic methods of COVID-19, detecting SARS-CoV-2 in collected samples. We reviewed the gold standard RT-qPCR method for COVID-19 diagnosis to confer a better understanding and application to combat the COVID-19 pandemic. This comprehensive review may further develop awareness about the management of the COVID-19 pandemic.

## 1. Introduction

Coronavirus disease 2019 (COVID-19) is a highly contagious communicable disease of the present time caused by a novel coronavirus, severe acute respiratory syndrome coronavirus-2 (SARS-CoV-2) [1]. It is believed that this viral infection was initiated with a zoonotic transfer from a seafood market in Wuhan, China [2]. Initially, the viral outbreak was considered endemic in China but, within a few weeks, the SARS-CoV-2 infection causing COVID-19 was declared a global pandemic by the World Health Organization (WHO) on 11 March 2020 [3]. Till now, the virus has infected 535,863,950 individuals worldwide and is infecting new individuals consistently, developing new clusters of infection (https://covid19.who.int/, accessed on 16 June 2022). We have lost 6,314,972 people and persons aged 65 and older with compromised immunity and with underlying medical conditions, such as chronic lung or liver disease, asthma, diabetes, severe heart problems, etc., are at significant risk of illness, morbidity, and mortality (https://www.cdc.gov/coronavirus/2019-ncov/hcp/clinical-care/underlyingconditions.html, accessed on 16 June 2022). The detailed chronology and epidemiology of the virus are discussed elsewhere [4]. After the identification of SARS-CoV-2 as the etiological agent of the illness, a race against time was started to develop rapid and efficient diagnostic methods, opening a new avenue for diagnostic innovations [5]. With the availability of the viral genome sequence, quantitative polymerase chain reaction (qPCR) was rapidly adopted as a reliable test for the diagnosis of infection [6]. Although exponential new studies propose novel therapeutic interventions and vaccines, there is a knowledge gap for understanding COVID-19 pathogenesis thoroughly and devising effective strategies to combat the virus in an attempt to alleviate human suffering.

Despite efficient testing and tracing of the infected individuals being central to the countermeasures against the management of the COVID-19, inaccurate testing can undermine these measures against the spread of the infection [7]. Contrarily, a false-positive result can cause avoidable psychological distress, besides wasting resources to manage the nonpatient [8]. This review attempts to encapsulate the current knowledge of the viral pathophysiology with disease diagnosis and to critically analyze the reverse transcriptase-polymerase chain reaction (RT-qPCR) technique, one of the gold standards for the detection of SARS-CoV-2 infection [9,10,11,12,13].

## 2. History of SARS-CoV-2 or Epidemiology

SARS-CoV-2 belongs to the genus ß-coronavirus of the *coronaviridae* family of viruses [14,15]. This family comprises enveloped and positive-sense, linear, single-stranded RNA viruses [16]. Among RNA viruses, coronaviruses contain the largest genome [17]. The genome of the early isolate of SARS-CoV-2 from Wuhan is 29,903 nucleotides and the genome size of all other isolates is approximately 30 kb, which is typical of coronaviruses [18,19]. A distinguishing feature of these viruses is the presence of spike-like projections on the surface, which appear like a crown under the electron microscope [11,16]. Hence, these viruses are named coronaviruses as “corona” means “crown” in Latin. These viruses can infect various animals, including humans, are spread through tiny respiratory droplets by direct or indirect contact with infected objects, and can cause respiratory illnesses like the common cold and severe acute respiratory syndrome [16,20]. Out of a total of seven, four human infections caused by coronavirus causing common cold and infection of the upper respiratory tract with mild symptoms are HKU1, NL63, OC43, and 229E [21,22]. These viruses spread via coughing and sneezing and cause mild upper respiratory illness in adults. HKU1, NL63, OC43, and 229E contribute to 15–30% of common cold cases in human adults but can cause severe life-threatening lower respiratory tract infections in immunocompromised individuals, infants, and older person [21]. Coronaviruses responsible for NL63 and 229E are believed to have originated from bat reservoirs, but OC43 and HKU1 are considered rodent-associated [23,24,25]. In addition to human coronaviruses, there are other coronaviruses that exclusively infect animals. Furthermore, the interspecies transmission of these animal viruses to human beings are an emerging threat to human health [26]. Among other animals, wild bats are considered a reservoir of coronaviruses because of their sequence similarity to human coronaviruses [26]. It was reported that SARS-CoV-2 is genetically similar to BatCoV RaTG13 (a BatCoV), indicating that bats might be the natural reservoir of SARS-CoV-2 [4].

In the past two decades, coronaviruses caused two epidemics, SARS-CoV [27,28] and the Middle East respiratory syndrome (MERS) [29]. In February 2003, severe acute respiratory syndrome (SARS) originated in China. The virus spread from China to Hong Kong and to other Asian countries [30]. The spread of SARS proclaimed a new and efficient medium of transmission of viruses by international air travel. The disease symptoms were fever, cough, chest pain, dyspnea, and hypoxemia (low blood oxygen levels), and the global fatality rate was 11% [30,31,32]. There was no available treatment or vaccine against SARS, and the control of the outbreak has relied completely on precaution, detection, and surveillance. Significant steps toward this were disinfection of aircraft and cruise vessels, identification of patients, isolation of patients, contact tracing, and quarantine of symptomatic and asymptomatic persons having any contact with SARS-infected individuals. Additionally, social distancing, wearing masks, frequent handwashing with soap, and use of alcohol-based sanitizer were significant steps in controlling the disease [30,33]. In 2012, MERS was identified in several countries like Saudi Arabia, the United Arab Emirates, and Korea [34]. Symptoms of MERS were similar to SARS-CoV and included fever, cough, and dyspnea, sometimes accompanied by pneumonia and gastrointestinal problems [29]. According to WHO, the fatality rate that was due to MERS was approximately 35% and the lesson learned from SARS proved helpful in controlling MERS. All the precautionary measures of SARS were considered, in addition to prohibiting direct or indirect contact with dromedary camels, as camels were the primary host of MERS [35].

These SARS and MERS infections foreboded a more challenging situation and an upcoming menace. COVID-19 is the third severe disease caused by coronaviruses and poses a severe threat to the world economy and public health. On 11 February 2020, the International Committee on Taxonomy of Viruses (ICTV) named the etiologic agent of COVID-19 as SARS-CoV-2, which was previously known as the 2019 novel coronavirus [15]. This name was proposed because of the higher homology of this new coronavirus with SARS-CoV, the causative agent of SARS [15]. Although SARS-CoV and SARS-CoV-2 are related, these two viruses possess differences. Unlike common-cold-causing human coronaviruses (229E, NL63, OC43, and HKU1) where the infection is confined to the upper respiratory tract, SARS-CoV, MERS-CoV, and SARS-CoV-2 spread from the upper respiratory tract and cause severe infection in the lower respiratory tract, leading to acute lung injury (ALI) and multi-organ failure, eliciting fatal outcomes [36]. Another noteworthy similarity between SARS-CoV and SARS-CoV-2 is the use of ACE2 as a receptor for entry into the host cell [37]. Human coronaviruses (hCoVs) are a constant threat to human health because of their emergence and reemergence, as is evident with SARS-CoV, MERS-CoV, and SARS-CoV-2 infections. Despite similarities between these viruses, there exist obvious differences, as a lesson learned from SARS-CoV is helpful in managing and containing MERS-CoV and SARS-CoV-2. The major problem with the current COVID-19 is the worldwide panic associated with the fast spread of misinformation causing an annoying infodemic [38]. Among these hCoVs, there is a difference in genome size; MERS-CoV possesses the largest genome with approximately 30.11 kb followed by SARS-CoV-2 (~29.9 kb) and SARS-CoV (29.75 kb) [39]. A better understanding of the genomes of hCoVs promotes combat against disease outbreaks by devising strategies for diagnostic systems, drug, and vaccine development [40].

## 3. Molecular Biology of SARS-CoV-2

SARS-CoV-2 is enveloped in a positive-sense single-stranded RNA virus with a genome size of 29,903 nucleotides (Figure 1) [41,42]. The virion size of this virus varies from 80–120 nm in diameter [43,44]. The nucleotide sequence of SARS-CoV-2 is 79.5% identical to SARS-CoV and 51.8% identical to MERS-CoV [41,45]. This suggests SARS-CoV-2 is closer to SARS-CoV. SARS-CoV and SARS-CoV-2 have similar lengths for most of the proteins. SARS-CoV-2 encodes four structural genes: spike glycoprotein (S), membrane glycoprotein (M), envelope glycoprotein (E), and nucleocapsid (N). The amino acid sequences of these structural genes are ~90% identical with SARS-CoV except the S gene [4,41]. The S protein of SARS-CoV-2 plays a crucial role in the viral entry into the host cell by binding to the host cell-surface receptor angiotensin-converting enzyme-2 (ACE2), and modifications in this protein may lead to different mechanisms and differential intensity of entry into the host cells [46,47,48]. Most of the SARS-CoV-2 non-structural proteins have greater than 85% amino acid sequence identity with SARS-CoV [49]. SARS-CoV-2 possesses four structural proteins: spike glycoprotein (S, 1273 amino acids), envelope glycoprotein (E, 75 amino acids), membrane protein (M, 222 amino acids), and nucleocapsid (N, 419 amino acids) (Figure 1) [41,50]. The N protein is involved in the RNA binding and packaging [50,51]. The most abundant protein in the outer membrane is M-glycoprotein. M and E proteins play a role in viral packaging and the S proteins play a crucial role in host cell binding and infection.

Entry of SARS-CoV-2 into host cells is mediated by binding the receptor-binding domain (RBD) of the S protein to host cell receptors ACE2 and TMPRSS2, a serine protease that helps in the primming of the S protein [52,53]. ACE2 is present in the lung on pneumocytes II, indicating the lung as the primary target organ of SARS-CoV-2. In addition to this, ACE2 also catalyzes the conversion of regulatory peptides in the cardiovascular system, responding to maintain the homeostatic state, and this activity may account for the rationale behind fatal symptoms including pulmonary embolism or deep venous thrombosis in severe COVID-19 patients [54]. However, the factors contributing to enhanced SARS-CoV-2 transmission are the efficient use of TMPRSS2 compared to SARS-CoV and the higher affinity for ACE2 owing to the modifications in the RBD leading to stabilizing virus-binding hotspots [44,52]. In addition, SARS-CoV-2 entry requires sequential cleavage of the spike glycoprotein at the S1/S2 and the S2’ cleavage sites to mediate membrane fusion. SARS-CoV-2 has a polybasic insertion (PRRAR) at the S1/S2 cleavage site that can be cleaved by furin (furin is a host-cell enzyme in human organs, such as the liver, the lungs, and the small intestines). These factors provide a mechanism called a spring-loaded manner of entry into the host cell, which prohibits endosomal trapping and is accountable for the higher transmissibility of SARS-CoV-2 compared to SARS-CoV [55]. The coronavirus spike (S) glycoprotein is a crucial target for vaccines, therapeutic antibodies, and diagnostics. The SARS-CoV-2 variants of concern (VOCs), Alpha, Delta, and Omicron, have mutations in the S1 subunit of the spike protein, which hosts the RBDs, hence altering the interaction of RBD with host-cell receptor ACE2, resulting in viral entry efficiency into the host cell (Table 1). The Alpha variant has ten modifications in the spike-protein sequence, which results in RBDs being more likely to stay in the ‘up’ position [56].

The Delta variant hosts multiple mutations in the S1 subunit, including three in the RBD that seem to improve the RBD’s ability to bind to ACE2 and evade the immune system [89]. These multiple mutations in spike proteins enable increased transmission and possible antibody resistance. These variants of SARS-CoV-2 tend to have alterations in furin cleavage sites. In both variants, proline at the 681 position is replaced with other amino acids: in the Alpha, variant proline has been replaced by histidine (P681H), while in the Delta variant, an arginine (P681R) has replaced the proline. These mutations help the virus to transmit into host cells more efficiently. The new Omicron variant has many modifications in the spike protein [90]. Preliminary data indicate that the patients with Omicron infection have mild symptoms, but there is an increased risk of reinfection [91].

## 4. Diagnostics for COVID-19

Depending on an individual’s age, immune responses, and associated co-morbidities, infection by SARS-CoV-2 leads to highly amassed responses in different individuals ranging from asymptomatic to individuals exhibiting enormously diversified symptoms. Young and healthy people show no or mild symptoms, but they may act as silent carriers and can cause covert infections [92]. Severe COVID-19 cases can end in hospitalization, some necessitating assisted mechanical ventilation, and some cases may be fatal [93].

Identifying infected individuals and asymptomatic viral carriers with rapid and accurate testing has played a pivotal role in containing and mitigating the COVID-19 pandemic. Identification of individuals infected with SARS-CoV-2, either symptomatic or asymptomatic, has prevented further person-to-person disease transmission (https://www.cdc.gov/coronavirus/2019-ncov/hcp/testing-overview.html, accessed on 16 June 2022). A coalition of multiple methods is in use to diagnose the presence of viral infection in individuals [94]. The primary steps for COVID-19 diagnosis are examining the presence of classical signs and symptoms such as fever or chills, cough, shortness of breath, muscle or body aches, headache, fatigue, sore throat, the new loss of taste or smell, dyspnoea, congestion, or runny nose, nausea or vomiting, conjunctivitis, and gastrointestinal issues (Figure 2) [6]. Furthermore, physical examination of signs including bronchial breath sounds, bronchophony, egophony, wheezing, crackles, rhonchi, and tests such as the anion gap blood test for respiratory acidosis or alkalosis and a complete blood count (CBC) to monitor thrombocytopenia and lymphopenia [95].

SARS-CoV-2 enters the human body as respiratory aerosols; samples from the oropharyngeal or nasopharyngeal are primarily used for viral detection. This virus travels from the upper respiratory tract to the lower respiratory tract, where viral replication occurs. Primarily, the upper respiratory system samples such as oropharyngeal swabs (OPS) and nasopharyngeal swabs (NPSs) are in use for COVID-19 diagnosis [96,97]. Other samples such as saliva, bronchoalveolar lavage (BAL), pleural fluid, tracheal aspirates, blood, urine, and fecal material can also be used for the detection of SARS-CoV-2 infection. For the monitoring and prognosis of the disease at every stage, effective diagnostic tests play a pivotal role. Since the initial report of the SARS-CoV-2 infection, numerous assay kits and tests have been developed for the purpose of COVID-19 diagnosis. Predominantly, there are two types of diagnostic methods in use: the first category is molecular genetics-based (viral test) and the second is serological-based (antibody test) (Figure 2). Among these reverse-transcriptase PCR, isothermal nucleic acid amplification, hybridization microarray assay, serological/immunological SARS-CoV-2 antibody ELISA, and chest CT are promising. In Table 2, a list of different diagnosis methods in use for COVID-19 diagnosis is provided. Advanced molecular biology techniques using polymerase chain reaction (PCR) in real time is a rapid testing method for SARS-CoV-2 infection. This technique is convenient and in use owing to the availability of the genome sequence of SARS-CoV-2. Adapting the PCR technique for COVID-19 diagnosis was straightforward as this technique is in use for the diagnosis of several other diseases, including previous coronavirus infections [11]. The following section describes in detail the use of gold standard RT-qPCR methods to detect the presence of SARS-CoV-2 in collected samples.

### 4.1. Reverse-Transcriptase PCR (RT-qPCR)

The nucleic acid amplification test (NAAT) by RT-qPCR is a sensitive, accurate, and globally accepted gold standard diagnostic method for the SARS-CoV-2 detection [9,10,108]. PCR is being used as a diagnostic test to detect pathogens, novel infections, and antimicrobial resistance profiling [11,109]. PCR is a precise and sensitive method to detect nucleic acids and possesses the potential to generate billions of copies of target DNA from a single copy [109]. This technique relied on an enzyme-driven process for amplifying short regions of DNA in vitro. The requirement of this method is information on at least partial sequences of the target DNA for designing oligonucleotide primers that hybridize specifically to the target sequences [109]. In clinical settings, real-time RT-qPCR is a revolutionary advancement where detection and expression analysis of gene(s) can be carried out in real time, as PCR reaction progresses, and amplification and analysis are done simultaneously in a closed system. This closed system further helps to minimize false-positive results associated with the amplification product contamination [110]. In addition to this, RT-qPCR is fast, sensitive, and reproducible; with the use of automated instrumentation, these features are further enhanced. Recently, NAAT have included other techniques such as isothermal amplification platforms with nicking endonuclease amplification reaction (NEAR), loop-mediated isothermal amplification (LAMP), and transcription-mediated amplification (TMA) [111]. A detailed overview of the RT-qPCR method for SARS-CoV-2 detection is depicted in Figure 3.

### 4.2. Specimens for Detection of SARS-CoV-2

The genetic material of SARS-CoV-2 (RNA) is first converted into complementary DNA (cDNA) by the action of RNA-dependent DNA polymerase (reverse transcriptase) prior to the actual amplification. For this, viral RNA can be collected from diverse specimens such as ocular secretions, saliva, sputum, bronchoalveolar lavage (BAL), blood, and fecal material, but upper respiratory system samples such as oropharyngeal swabs (OPS) and nasopharyngeal swabs (NPSs) are widely in use [96,97]. In detecting SARS-CoV-2 in various samples, limit of detection (LoD) plays a crucial role [112]. Presently, the best-of-class assay has LoD of ~100 copies of viral RNA per milliliters of transport media; assays with higher LoDs may result in a false negative [112]. Though OPS and NPSs are primarily in use because of lower LoDs, there is a recommendation for the use of combined swabs for COVID-19 diagnosis to avoid false-negative results [113]. Saliva has also been used as a reliable, noninvasive approach for SARS-CoV-2 detection and disease progression [114]. The advantages of using saliva for diagnosis are self-collection, reduced transmission risk during the sample collection, and also a lesser requirement of PPE, trained healthcare professionals, transportation, and storage costs [115]. Importantly, viral load over the course of the infection is detrimental to the analytical sensitivity of assays. It was reported in several studies that the viral load of SARS-CoV-2 peaks during or even shortly before the onset of symptoms and decreases rapidly within the first seven days [115,116]. Furthermore, the virus can be detected in samples for longer periods from the onset of symptoms, usually for 20 days or longer in some patients [117]. There are specific guidelines for sample collection for different specimens by the CDC (https://www.cdc.gov/coronavirus/2019-ncov/lab/guidelines-clinical-specimens.html, accessed on 16 June 2022). For NPSs and OPS, collecting using only synthetic fiber swabs with thin plastic or wire shafts specifically designed for sampling nasopharyngeal mucosa is recommended. For this patient, the head needs to be tilted back 70 degrees and the swab needs to be inserted slowly into the nostril to contact the nasopharynx. Thereafter, gently rub and roll the swab and leave it for a few seconds to absorb secretions; remove it slowly and place it in the transport tube. These samples can be stored at 2–8 °C for up to 72 h; for longer duration, samples must be stored at −70 °C. Extracted nucleic acid samples must be stored at −70 °C or lower. The collected specimen must be transported to the laboratory while maintaining a cold chain of 2–4 °C throughout [118].

### 4.3. Biomarkers/Genes Used for RT-qPCR

According to Centers for Disease Control and Prevention (CDC) and WHO guidelines, the RNA samples are reverse-transcribed into cDNA using different primers specific for the open reading frame 1ab (ORF1ab), ORF8, RNA-dependent RNA polymerase (RdRp), hemagglutinin-esterase (HE), and the nucleocapsid genes N1, N2, envelope genes (E), spike genes (S), and transmembrane gene (M), while human RNase P is used as control (Appendix A). Some other controls in use for each reaction are no template control, 2019-nCoV positive control, and human specimen control (CDC 2020) [119,120,121]. Additionally, ORF1ab and RdRp are included in RT-qPCR reactions to rule out any potential cross-reactivity, which may occur with other coronaviruses, and to avoid chances of genetic drift in the SARS-CoV-2 genome [122]. As per the CDC recommendation, screening must be done targeting nucleocapsid genes (N1 and N2), but the WHO recommendations require targeting E genes, which must be followed by confirmation using the RdRp gene [122]. Though there is less impact on the detection of SARS-CoV-2 because of emergent variants as most mutations accumulated in the S gene and not in other genes, which are a common target for detection assays. Some VOCs of SARS-CoV-2 (Alfa and Omicron) provides negative results or weaker signals with S-gene RT-qPCR assays, while positive ones with other genes (https://www.ecdc.europa.eu/sites/default/files/documents/Methods-for-the-detection-and-characterisation-of-SARS-CoV-2-variants-first-update.pdf, accessed on 16 June 2022). This effect of no detection of the S gene or weaker signals is referred as S-gene target failure (SGTF) and is due to deletion at nt207–212 (Δ69–70) [123]. Alfa and the majority of Omicron variants of SARS-CoV-2 give negative RT-qPCR results using the S gene, but positive ones with ORF1 and the N gene [124].

The RT-qPCR reaction can be performed in either one or two steps [125,126]. In the conventional two-step RT-qPCR, the reactions for cDNA synthesis and amplification of DNA are conducted separately in two sequential steps, while in one-step RT-qPCR, both the above-mentioned cDNA synthesis and DNA amplification reactions are performed in a single step within one tube containing the requirements to accomplish the entire assay [125]. In detecting SARS-CoV-2 for COVID-19 diagnosis, this one-step RT-qPCR is preferred over the two-step method owing to it being fast and efficient and involving limited sample handling, minimal experimental errors, and a reduced bench time [97,125]. This is followed by cDNA being amplified using fluorescent-based quantitative PCR assays to allow sensitive detection and quantification of the viral RNA [97]. Figure 4 shows the mechanistic steps of DNA amplification and its detection. The qPCR reaction steps are similar to the PCR steps, with initial denaturation of the template at 95 °C for 5–10 min followed by cyclic steps including denaturation (95 °C, 15–20 s), primer/probe annealing (60 °C, 15–20 s), and primer extension (72 °C, 1 min) for gene amplification. Annealing temperature plays a critical role in efficient amplification of the gene of interest and requires optimization and varies from template to template. The annealing temperature determines the qPCR efficiency and depends on the melting temperature (Tm) and is well-established for SARS-CoV-2 detection using different regions of the RNA genome. The qPCR is thereafter continued for 35–45 cycles; during each cycle, the template DNA amount is doubled, resulting in an increase in fluorescent signals. In Figure 4, the sigmoidal curve represents a typical result of the qPCR results, and this helps us interpret the assay outcomes. This curve has three distinct phases: up to cycle 15 or so the curve is near the baseline, in the second phase there is a strong upswing of the cure, usually between 15–30 cycles, and in this phase the amplification signal crosses the threshold. In the third phase, generally after 30 cycles there is a plateau where amplification tapers off and ceases to grow. This curve helps in determining the cycle threshold (Ct) value; this is the point where the curve first clearly rises off the baseline to a statistically significant degree. Crossing this noise threshold is the basis for calling a sample positive in the qualitative assay and the Ct value is the basis for the generation of the standard curve used in the quantifying template in quantitative PCR.

### 4.4. Reagents (Dyes)

In real-time RT-qPCR, the monitoring of amplification can be done in real time using fluorescent DNA-intercalating dyes such as SYBER green. This dye can bind nonspecifically to the double-stranded DNA generated during the amplification process [127]. There is a more popular alternative approach that uses a fluorescent-labeled internal DNA probe that specifically anneals within the target amplification region and a quencher molecule; this is the case with TaqMan assays [97]. In the TaqMan assay, a fluorescent-labeled oligonucleotide (short DNA molecule) probe is added that is labeled at both the 5′ and 3′ ends. In this, a fluorescent reporter is placed at the 5′ end of the probe and a quencher at the 3′ end, which is also fluorescently labeled. Until there is no amplification, both the 5′ reporter and 3′ quencher are in close proximity and no signal is detected. A fluorescent signal is detected only after the 5′ end reporter and the 3′ end quencher are separated (Figure 4). This separation of reporter and quencher usually takes place because of the enzymatic reaction during RT-qPCR, where the probe is incorporated into the PCR product. The TaqMan assay is more specific and sensitive as it depends upon two processes: first, the primer binding to its specific target sequences and, second, the probe binding to a specific complementary sequence in the downstream region of the primer [128]. An automated system further repeats the amplification process for up to approximately 40 cycles until the viral cDNA can be detected, usually by a fluorescent or electrical signal [129]. There is an effort for the rapid development of fully automated RT-qPCR methods and machines that can be used for quick, accurate results. There are high-throughput machines available that can be used to test 35,000 samples per day and this is further scalable up to 150,000 assays per day. The TaqMan RT-qPCR assay is considered highly sensitive and reproducible; hence, this method can produce reliable results [130]. Using two or more probes, real-time multiplex PCR can be performed to simultaneously detect multiple targets in a single reaction [131,132]. Figure 4 shows the mechanistic details of fluorescent probe-based real-time PCR.

### 4.5. Ct Value/Threshold Value

In the process of real-time PCR, the target genes are amplified and doubled with each cycle; thus, amplification occurs exponentially. As amplification proceeds, an increasing number of targets become available and the fluorescent signal increases exponentially, producing an exponential curve. The cycle threshold (Ct) value refers to the number of cycles of amplification required for the fluorescence signal of the PCR product or nucleic acid target to be detected or measurable and crossing a threshold or cut-off value is an indication of a positive RT-qPCR test result of a subjected sample [133,134]. This fluorescent signal intensity reflects the amounts of DNA amplicons present at the particular time; generally, after 30–35 cycles the viral cDNA can be quantified, even starting with a very small amount of viral RNA [126].

On the basis of internal controls, RT-qPCR tests can be either qualitative or quantitative, and this affects how a Ct value can be interpreted. In a qualitative RT-qPCR test, known amounts of virus are used to determine whether the Ct values are associated to determine positive and negative test results. In testing a specimen, a Ct value helps interpret a test result as positive or negative, but it cannot be used to determine the exact amount of virus present in an individual patient specimen. In a quantitative RT-qPCR test, a range of known numbers of genome copies (reference samples) are tested as a control in each RT-qPCR reaction; comparing the Ct value of a specimen to the Ct values from the reference samples, the test can calculate the copy number of target nucleic acid. The US Food and Drug Administration (FDA) emergency use authorization (EUA) has approved all SARS-CoV-2 RT-qPCR diagnostic kits only for qualitative test purposes (www.cdc.gov/coronavirus/2019, accessed on 16 June 2022). A list of available RT-qPCR kits for the detection of SARS-CoV-2 approved by the FDA under EUA is provided in the Appendix A.

The correlation between Ct value and viral load may be a useful tool for comparison purposes of certain populations including symptomatic and asymptomatic populations. Despite an association between the Ct value and the amount of genetic material in the tested samples, attempting to correlate Ct values and the amount of virus in the original specimen may be faulty. The Ct values of tested samples can be affected by various factors other than viral load, including but not limited to improper collection or storage, processing, or the sensitivity level of the test performed. Thus, a high Ct value can result from factors unrelated to the amount of virus present in the specimen. Hence, Ct values should not be used to infer a relationship with the viral load from a specimen, nor should they be used to determine the level of infection risk (https://www.cdc.gov/coronavirus/2019-ncov/lab/faqs.html, accessed on 16 June 2022). Some countries including India provide Ct values in RT-qPCR results. The significance of this Ct value is that it determines infectivity; a Ct value of 35 or lower is considered COVID-19 positive, while a patient with a Ct value higher than 35 is considered negative for COVID-19.

Additionally, concerns with the experimental result are false-positive or false-negative detections. A false-positive test refers to a false indication for infection present without any infection or presence of virus, while a false-negative test leads to patients declared to be “uninfected”, despite being infected [135]. The main reason for false-positive results is laboratory error, sample contamination, and cross-reactivity or off-target reactions (the test cross-reacting with something that is not SARS-CoV-2), while false-negative RT-PCR results can be due to a low level of viral RNA, improper sample collection, loss or damage during transportation, inefficient extraction, and improper storage conditions. Furthermore, positive PCR results indicate the presence of viral RNA, but this may not necessarily confirm the presence of the infectious virus. Finally, PCR positivity depends on specimen types; it declines more rapidly while using NP swabs compared to the sputum [136].

## 5. Limitations of RT-qPCR Detection Technique for SARS-CoV-2

Despite wide acceptance and use of the real-time PCR (qPCR) method as a gold standard molecular test of choice with high specificity and accuracy, it has limitations. This method demands professional skilled personnel and is associated with a high cost of instruments and a laboratory setup with a biosafety level 2 cabinet. This method requires absolute cleanliness and a sterile environment because of the high sensitivity of the assay, which can be contaminated easily and may sequalae in false-positive results. A false-positive result may occur because of contamination; furthermore, this can occur because of the presence of shedding of viral residual RNA in recovered patients. Furthermore, a false negative is the prime concern of many available commercial RT-qPCR kits because of lower diagnostic efficiency than optimal [137,138]. Considering the high incidence of false negative RT-qPCR results, the US Food and Drug Administration (FDA) has concluded that a negative RT-qPCR does not completely rule out the SARS-CoV-2 infection. To overcome the challenges of conventional RT-qPCR, many biomedical companies have developed diagnostic platforms that are fully automated and take less time to declare results [139].

## 6. Future Perspectives and Conclusions

COVID-19 is one of the deadliest pandemics in world history and requires the unmet attention of each and every citizen of the world to control the pandemic. The first step in containing SARS-CoV-2 is the detection of infected persons and appropriate isolation and treatment. Toward this aim, the need for efficient detection methods is imperative, with fast, accurate, and reliable result outputs. For this, RT-qPCR-based detection is the gold standard method of the present time and constant improvements for better and faster screening are in progress. Furthermore, asymptomatic individuals act as carriers and transmit the virus unknowingly. To stop this carrier-mediated viral transmission, all individuals must be tested for infection on a regular basis. Hence, the development of herd immunity using a vaccine regimen, discovery, and availability of suitable therapeutics and regular testing may help us fight against the COVID-19 pandemic.

## Figures and Tables

**Figure 1 diagnostics-12-01503-f001:**
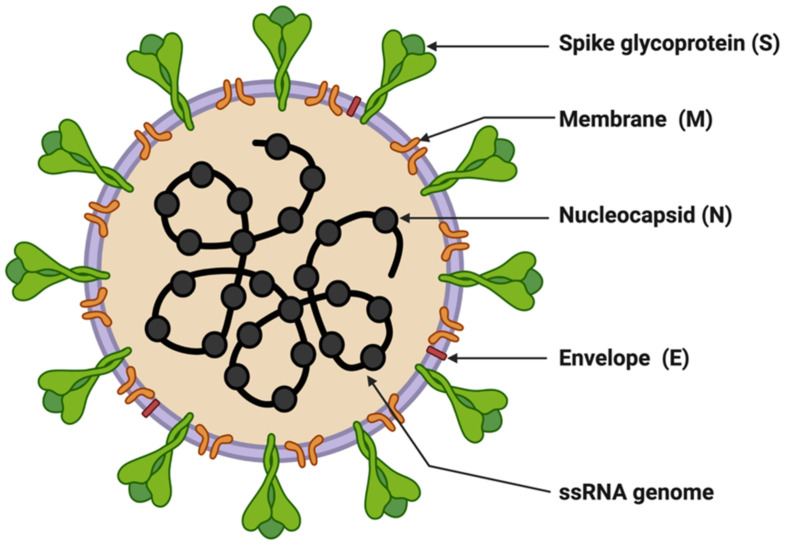
Structure of SARS-CoV-2. The figure was created with Biorender.com on 8 June 2022.

**Figure 2 diagnostics-12-01503-f002:**
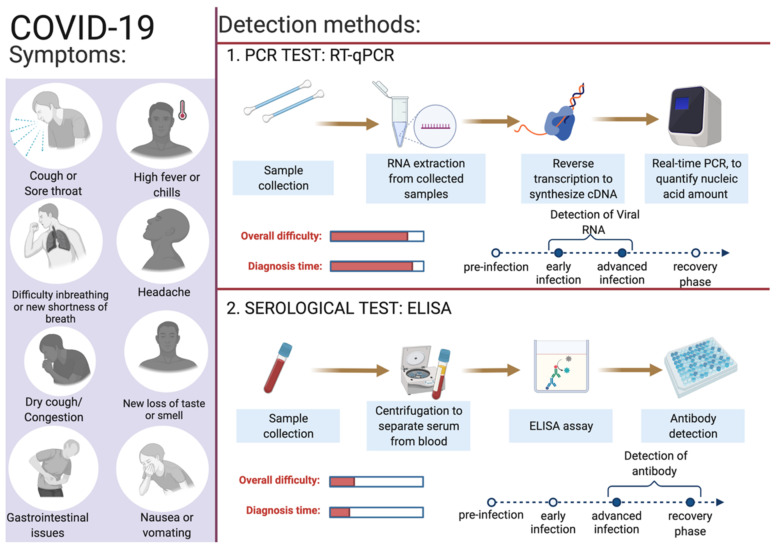
Overview of COVID-19 symptoms and SARS-CoV-2 detection methods for COVID-19 diagnosis. This figure was created with Biorender.com on 6 June 2022.

**Figure 3 diagnostics-12-01503-f003:**
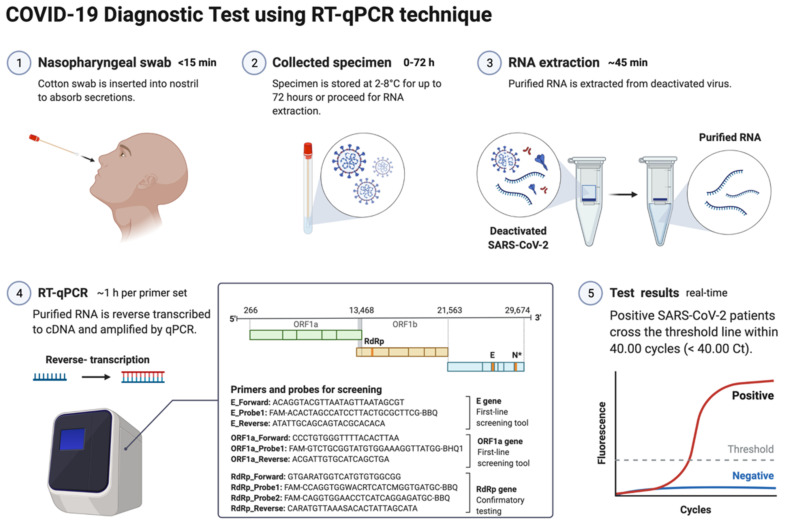
Schematic representation of COVID-19 diagnostic test using RT-PCR. This figure was created with Biorender.com on 18 May 2022.

**Figure 4 diagnostics-12-01503-f004:**
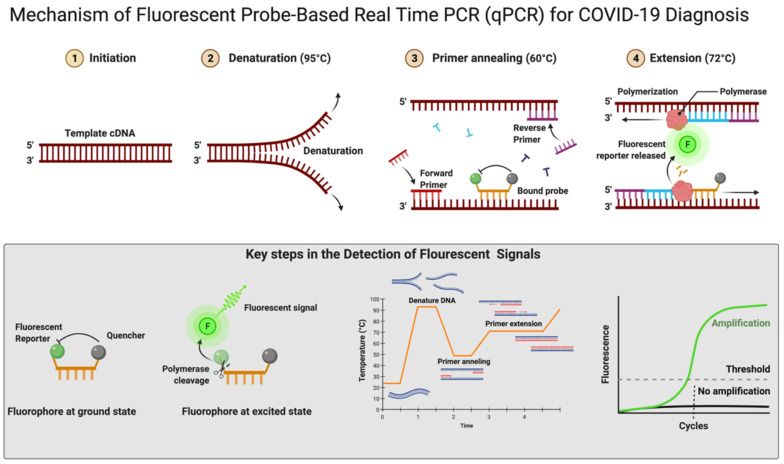
Mechanism of fluorescent probe-based real-time PCR (qPCR) for COVID-19 diagnosis. Figure was created with Biorender.com on 20 May 2022.

**Table 1 diagnostics-12-01503-t001:** List of SARS-CoV-2 variants.

S.No.	Name of Variant	Lineage	Earliest Sample	First Outbreak	Designated	Reference
1.	Epsilon	B.1.429, B.1.427	March 2020	United States	5 March 2021	[57,58]
2.	Zeta	P.2	April 2020	Brazil	17 March 2021	
3.	Beta	B.1.351	May 2020	South Africa	18 December 2020	[59,60]
4.	Lambda	C.37	August 2020	Peru	14 June 2021	[61,62,63]
5.	Alpha	B.1.1.7	September 2020	United Kingdom	18 December 2020	[64,65]
6.	Delta	B.1.617.2	October 2020	India	11 May 2021	[66,67]
7.	Gamma	P.1	November 2020	Brazil	11 January 2021	[68,69]
8.	Lota	B.1.526	November 2020	United States	24 March 2021	[70,71]
9.	Eta	B.1.525	December 2020	Multiple Countries	17 March 2021	[72,73]
10.	Kappa	B.1.617.1	December 2020	India	4 April 2021	[74,75]
11.	Theta	P.3	January 2021	Philippines	24 March 2021	[76]
12.	Mu	B.1.621	January 2021	Colombia	30 August 2021	[77,78]
13.	B.1.1.318	GR	January 2021	Multiple Countries	2 June 2021	[79]
14.	C.1.2	GR	June 2021	South Africa	1 September 2021	[80]
15.	B.1.640	GH/490R	September 2021	Multiple Countries	22 November 2021	[81]
16.	Omicron	BA.1	November 2021	South Africa	26 November 2021	[82,83]
17.	Omicron	BA.2	November 2021	South Africa	26 November 2021	[84,85]
18.	Omicron	BA.3	November 2021	South Africa	26 November 2021	[86]
19.	Omicron	BA.4	January 2022	South Africa	12 May 2022	[87]
20.	XD	Omicron BA.1 and Delta	January 2022	France	9 Mar, 2021	[88]
21.	Omicron	BA.5	February 2022	South Africa	12 May 2022	[87]

**Table 2 diagnostics-12-01503-t002:** List of different diagnostic methods in use.

Test	Technique	Specimen	Advantages	Disadvantages	Reference
Viral test (Molecular genetics based)					
Antigen	Lateral flow immunoluminescent assay, single or double target	NPS and ANS	Rapid, point-of-care tests	Less sensitive, and chances of false positives	[98]
Nucleic acid	RT–qPCR	Saliva, NPS, nasal mid-turbinate and ANS	Sensitive, specific	Expensive, requires laboratory personnel, specialized lab equipment and reagents	[98]
Nucleic acid	Loop-mediated isothermal amplification (LAMP)	Saliva, urine, NPS, nasal mid-turbinate and ANS	Sensitive, specific, rapid	Complicated designing of assay, chances of false positives	[99,100]
Nucleic acid	Recombinase polymerase amplification (RPA)	NPS and ANS	Sensitive, specific, rapid	Complicated designing of assay, expensive	[101]
Nucleic acid	Nicking endonuclease amplification reaction (NEAR)	NPS and ANS	Sensitive, rapid	Chances of false negatives	[100,101]
Nucleic acid	Transcription mediated amplification (TMA)	NPS and ANS	Sensitive, specific	Expensive and less flexible	[102]
Nucleic acid	Helicase-dependent amplification (HDA)	NPS and ANS	Sensitive, rapid	Chances of false positives	[100]
Nucleic acid	Clustered regularly interspaced short palindromic repeats (CRISPR)	AN, OPl, NP wash/aspirate and BAL	Sensitive, specific, rapid, versatile	Target sequences of the Cas proteins are restricted; multiplexing can create interferences which may lead to cross-reactivities	[100,103]
Nucleic acid	Strand displacement amplification (SDA)	NPS and ANS	Rapid, sensitive	Reverse transcription of virus RNA is required, shortcomings of chosen isothermal method.	[104]
Volatile organic compounds (VOCs)	Rapid gas chromatography-mass spectrometry (GC-MS)	Breath	Rapid	Presumptive	[98]
Radiological abnormalities caused by viral infection	Computed Tomography	Cross-sectional images of patient’s chest	Non-invasive, lesser expensive	Less specific because imaging features overlap with other viral pneumonia	[105]
Serological/Immunological test					
Antibody	Enzyme-linked immunosorbent assay (ELISA) and chemiluminescent immunoassay (CIA)	Blood and tissue specimens	Rapid, point-of-care tests, can identify previous infection	Dependent on duration of infection, false-negative results	[106]
Antibody	Dried blood spot (DBS)	Dried blood samples pricked from fingers	Sensitive and rapid	Storage temperature sensitive	[107]

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
