# Peer review of "COVID-19 Diagnosis: A Comprehensive Review of the RT-qPCR Method for Detection of SARS-CoV-2"

_diagnostics, 2022, doi:10.3390/diagnostics12061503_

Round 1
Reviewer 1 Report
Comments to the Author
The manuscript titled with “COVID-19 diagnosis: A comprehensive review of the RT-PCR 2 method for detection of SARS-CoV-2 topic itself is interesting and is suitable for this journal. In this study, however, there are several key remarks when it comes to the coverage and explanation.
The authors did not clearly explain in the given below paragraph:
- History of SARS-CoV-2 or Epidemiology
- a. Genome size of the SARS-CoV-2?
- b. Structure of SARS-CoV-2 should be in this section?
- c. Add reference “Identification of a new human corona…….” Lia Vander et al, Nature medicine 2004
- d. In line no. 81 mention the reference together of all the symptoms like HKU1, NL63 etc.
- e. Author has not explained about the Corona virus of Bat? Ref. “Identification of a novel coronavirus…..” LLM Poon et al ASM 2005
- f. Mention literature about the virus structure and protein Ref. “Structure, function and evolution….” Fang Li, Annual review of virology 2016
- g. Author could not explain the difference between the SARS, MERS and Corona virus?
Ref. “Covid 19……” Zhixing Zhu, BMC 2020
- h. Author can contribute in details about the meaning of the name of SARS-CoV-2?
- Molecular Biology of SARS-CoV-2
- a. Author can discuss only about the SARS-CoV-2. In this section no need of the information about the MERS.
- b. Author can contribute more about the molecular biology of SARS-CoV-2.
Ref. “An overview of……………” Ankur Das et al, Gene reports 2021.
- c. Author can make a box for the information of different variant of SARS-CoV-2.
- Diagnostics for COVID-19
- a. Author can cite WHO proceedings “Methods for the detection and identification………..” March 2021.
- b. Author most add a list of different diagnosis methods which is currently use for the detection SARS-CoV-2?
- c. Author couldn’t explain the detection limit of the SARS-CoV-2 in different samples?
- d. Author couldn’t explain the different diagnostic methods of the SARS-CoV-2 like serological test etc?
- a. Reverse-Transcriptase PCR (RT-PCR)
- a. In figure 1, shift structure of SARS-CoV-2 in the paragraph 1 (history……).
- b. In figure 1, Author can add more symptoms images and shift in the other proper section.
- c. Line no. 198 not relevant in this relevant with this section.
- d. In figure 2, add primer of the other gene also like S, ORF1a etc.
- b. Specimens for detection of SARS-CoV-2
- a. By the author it will be explain sample collection method, storage condition of sample and sample transportation method.
- c. Biomarkers/genes used for RT-PCR
- a. Table 1 is missing
- b. Author should add about the different variant of SARS-CoV-2 gene and their detection limitation
- c. In figure 3, author mention about the denaturation, annealing and extension temperature selection and not explain the criteria of its selection in the text.
- d. In figure 3, author does not explain the sigmoidal curve of the RT- PCR result.
- d. Reagents (dyes)
- a. Author can mention the name of the intercalating dye
- b. In this section, information is very superficial. I suggest to re-write.
- c. Author not contributed about the different RT-PCR reagents and their uses.
- e. Ct Value/Threshold value
- a. Author can contribute about the significance of the Ct value.
- Limitations of RT-PCR detection technique for SARS-CoV-2
- a. Author can not explained the reason of the false positive results?
- b. Author mentioned in line no. 323 “takes 2-5 days from sample collection to result declaration” now scenario is changed globally. rechecking is required.
Other major comments:
- Author does not mention about the RNA extraction method.
- Author does not enlighten on the manual and fully automatic methods.
- Result interpretation part is missing.
- The manuscript is written very superficially in most of the section. It is important to mention some more information or deep information about the test methods.
Author Response
Comments to the Author
The manuscript titled with “COVID-19 diagnosis: A comprehensive review of the RT-PCR 2 method for detection of SARS-CoV-2 topic itself is interesting and is suitable for this journal. In this study, however, there are several key remarks when it comes to the coverage and explanation.
Response: We appreciate the effort of the reviewer in providing suggestions to strengthen the manuscript. We have modified the manuscript, following are the pointwise response which has been incorporated into the manuscript with track changes.
The authors did not clearly explain in the given below paragraph:
- History of SARS-CoV-2 or Epidemiology
- a. Genome size of the SARS-CoV-2?
Response: The initial isolate of SARS-CoV-2 from Wuhan, China has a 29903 nt ≈ 30 kb ssRNA genome (NCBI MN908947.3), This information is now incorporated in the manuscript and reference for this is also added.
- b. Structure of SARS-CoV-2 should be in this section?
Response: Yes, this is a brief overview, and without this information, it will not be easy to follow the subject. However, a detailed information about structure is now added to the section 2. Molecular biology of SARS-CoV-2, also a figure depicting structure of SARS-CoV-2 is included as figure 1 to this section as advised by reviewer elsewhere.
- c. Add reference “Identification of a new human corona…….” Lia Vander et al, Nature medicine 2004
Response: Reference added.
- d. In line no. 81 mention the reference together of all the symptoms like HKU1, NL63 etc.
Response: Now this is modified as advised by reviewer.
- e. Author has not explained about the Corona virus of Bat? Ref. “Identification of a novel coronavirus…..” LLM Poon et al ASM 2005
Response: This is now added, and reference is also incorporated as advised by the reviewer.
- f. Mention literature about the virus structure and protein Ref. “Structure, function and evolution….” Fang Li, Annual review of virology 2016
Response: This is now modified with relevant information about virus structure and protein and reference is also incorporated as advised by the reviewer.
- g. Author could not explain the difference between the SARS, MERS and Corona virus?
Ref. “Covid 19……” Zhixing Zhu, BMC 2020
Response: Difference between SARS, MERS and Corona virus is now incorporated in the manuscript with track changes for convenience to visualize by the reviewer. This reference is also incorporated as advised by the reviewer.
- h. Author can contribute in details about the meaning of the name of SARS-CoV-2?
Response: ICTV announced “severe acute respiratory syndrome coronavirus 2 (SARS-CoV- 2)” as the name of the new virus on 11 February 2020. This name was chosen because the virus is genetically related to the coronavirus responsible for the SARS outbreak of 2003. While related, the two viruses are different. This information is now incorporated to the manuscript with track changes.
- Molecular Biology of SARS-CoV-2
- a. Author can discuss only about the SARS-CoV-2. In this section no need of the information about the MERS.
Response: Considering the reviewer's suggestion, information is confined to SARS-CoV-2, other information has been removed.
- b. Author can contribute more about the molecular biology of SARS-CoV-2.
Ref. “An overview of……………” Ankur Das et al, Gene reports 2021.
Response: We have now included more information and the above-mentioned reference is also cited in the manuscript.
- c. Author can make a box for the information of different variant of SARS-CoV-2.
Response: A table is now added (Table 1) with pieces of information on SARS-CoV-2 variants.
- Diagnostics for COVID-19
- a. Author can cite WHO proceedings “Methods for the detection and identification………..” March 2021.
Response: Now added to the manuscript.
- b. Author most add a list of different diagnosis methods which is currently use for the detection SARS-CoV-2?
Response: Now a list of different diagnosis methods which is currently use for the detection SARS-CoV-2 added to the manuscript as Table. 2.
- c. Author couldn’t explain the detection limit of the SARS-CoV-2 in different samples?
Response: Now the detection limit of the SARS-CoV-2 in different samples is discussed in the manuscript.
- d. Author couldn’t explain the different diagnostic methods of the SARS-CoV-2 like serological test etc?
Response: Now different diagnostic methods of the SARS-CoV-2 like serological test etc are discussed in the manuscript with track changes, visible in figure 2 and list of different diagnosis methods which is currently use for the detection SARS-CoV-2 added to the manuscript as Table. 2.
- a. Reverse-Transcriptase PCR (RT-PCR)
- a. In figure 1, shift structure of SARS-CoV-2 in the paragraph 1 (history……).
Response: a figure depicting structure of SARS-CoV-2 is included as new figure 1.
- b. In figure 1, Author can add more symptoms images and shift in the other proper section.
Response: This is now modified, and symptoms are added and provided as figure 2.
- c. Line no. 198 not relevant in this relevant with this section.
Response: Following the advice of the reviewer now this is removed.
- d. In figure 2, add primer of the other gene also like S, ORF1a etc.
Response: Now this is modified and primer sequence of ORF1 is added, this now figure 3. The information in this figure is an example of PCR primers and probes for RT-qPCR. To keep the cleanliness of figure primer and probes for other target genes are not included, this is for an example purpose and these information’s are easily available.
- Specimens for detection of SARS-CoV-2
- a. By the author it will be explain sample collection method, storage condition of sample and sample transportation method.
Response: Following reviewer advice now we have explained in detailed about sample collection methods, storage conditions and sample transportation methods. These information’s are incorporated to the manuscript with track changes.
- c. Biomarkers/genes used for RT-PCR
- a. Table 1 is missing
Response: Table 1 is now incorporated as modified Table S 1 at the end of the manuscript.
- b. Author should add about the different variant of SARS-CoV-2 gene and their detection limitation
Response: Information about different variant of SARS-CoV-2 gene and their detection limitation is discussed in the manuscript.
- c. In figure 3, author mention about the denaturation, annealing and extension temperature selection and not explain the criteria of its selection in the text.
Response: Now selection criteria for these denaturation, annealing and extension temperature is discussed in the manuscript.
- d. In figure 3, author does not explain the sigmoidal curve of the RT- PCR result.
Response: This is now figure 4 and we have discussed about the interpretation about sigmoidal curve of RT-qPCR results in the manuscript.
- d. Reagents (dyes)
- a. Author can mention the name of the intercalating dye
Response: This is now mentioned in the manuscript.
- b. In this section, information is very superficial. I suggest to re-write.
Response: A portion of this section is now rewritten with relevant information.
- c. Author not contributed about the different RT-PCR reagents and their uses.
Response: This is now mentioned and explained about the different RT-PCR reagents and their uses.
- e. Ct Value/Threshold value
- a. Author can contribute about the significance of the Ct value.
Response: Significance about Ct values are now provided in the manuscript.
- Limitations of RT-PCR detection technique for SARS-CoV-2
- a. Author can not explained the reason of the false positive results?
Response: Now this section is modified with proper information for false-positive results.
- b. Author mentioned in line no. 323 “takes 2-5 days from sample collection to result declaration” now scenario is changed globally. rechecking is required.
Response: We rechecked this statement and is now changed.
Other major comments:
- Author does not mention about the RNA extraction method.
Response: This information is common and not require for this manuscript, this is an overview and not a protocol manuscript or research article. We have included some methods about sample collection, storage, and transportation. I hope reviewer will agree with our justification as this is not relevant for the present manuscript.
- Author does not enlighten on the manual and fully automatic methods.
Response: Now this is discussed in the manuscript.
- Result interpretation part is missing.
Response: Now this is discussed in the manuscript.
- The manuscript is written very superficially in most of the section. It is important to mention some more information or deep information about the test methods.
Response: Now this is addressed following reviewers’ suggestion and discussed in detailed as suggested in the manuscript.
Submission Date
20 April 2022
Date of this review
29 Apr 2022 00:05:49
Reviewer 2 Report
The work entitled “COVID-19 diagnosis: A comprehensive review of the RT-PCR method for detection of SARS-CoV-2” is a review that mainly characterizes the PCR technique as a method of diagnosis of SARS-CoV-2 infection. It is not innovative and does not contain any new conclusions from the literature review. Nevertheless, it provides a good quick overview for readers who are very distant about the topic. I believe that the work presented for review could be published after corrections.
Major revision:
- Line 323 "takes 2-5 days from sample collection to result declaration". This sentence isn’t true. The time to obtain the result from the sample collection is 3-4 hours. Certainly not days.
- Line 320-321 "This method is labor-intensive". This is definitely not a laborious method.
- Line 318 - the entire chapter 4. It is a method that undoubtedly requires qualified personnel and equipment. It is also expensive compared to other detection methods. What is very important, it requires absolute cleanliness and sterile conditions, because it is a very sensitive method and easy to contaminate. The reaction product may also cause contamination and false-positive results.
- Line 329 "The COVID-19 pandemic is the deadliest in world history". Is that the truth?
- Line 193-194 "real-time Reverse-Transcriptase-PCR (RT-PCR)". Incorrect name. We write real-time PCR or quantitative PCR, the correct abbreviation is RT-qPCR. We never develop RT - Reverse-Transcriptase. Please apply to all.
- Line 215 - figure 2 replace with Figure 2.
- Chapter 3 - in subsections, replace letters with numbers, for example, 3.1; 3.2. According to the instructions for authors.
Author Response
Reviewer 2
Comments and Suggestions for Authors
The work entitled “COVID-19 diagnosis: A comprehensive review of the RT-PCR method for detection of SARS-CoV-2” is a review that mainly characterizes the PCR technique as a method of diagnosis of SARS-CoV-2 infection. It is not innovative and does not contain any new conclusions from the literature review. Nevertheless, it provides a good quick overview for readers who are very distant about the topic. I believe that the work presented for review could be published after corrections.
Response: We are thankful to the reviewer for the precious time spent on reviewing our manuscript and for providing us with valuable suggestions. We sincerely agree with the reviewer’s comment as this could be a good quick overview for readers on the topic. We attempted to provide comprehensive literature and an easy-to-understand process of COVID-19 detection method using RT-qPCR. Below is the pointwise response to each comment and suggestions are incorporated into the manuscript with track changes.
Major revision:
- Line 323 "takes 2-5 days from sample collection to result declaration". This sentence isn’t true. The time to obtain the result from the sample collection is 3-4 hours. Certainly not days.
Response: We appreciate the reviewer’s minute observation, the process takes around 3-4 hours from sample collection to result declaration, not 2-5 days. However, this rapid detection technique is limited due to delayed transportation of samples from the collection site to the laboratory as well as the requirement of a large number of samples to batch samples in a large run.
- Line 320-321 "This method is labor-intensive". This is definitely not a laborious method.
Response: We agree with the reviewer’s comment and now deleted the term “labor-intensive”.
- Line 318 - the entire chapter 4. It is a method that undoubtedly requires qualified personnel and equipment. It is also expensive compared to other detection methods. What is very important, it requires absolute cleanliness and sterile conditions, because it is a very sensitive method and easy to contaminate. The reaction product may also cause contamination and false-positive results.
Response: Chapter 4 modified.
- Line 329 "The COVID-19 pandemic is the deadliest in world history". Is that the truth?
Response: Line 329 "The COVID-19 pandemic is the deadliest in world history" is now changed to "The COVID-19 pandemic is one of the deadliest pandemics in world history". This is now more appropriate to consider.
- Line 193-194 "real-time Reverse-Transcriptase-PCR (RT-PCR)". Incorrect name. We write real-time PCR or quantitative PCR, the correct abbreviation is RT-qPCR. We never develop RT - Reverse-Transcriptase. Please apply to all.
Response: We appreciate the suggestion and changed it to RT-qPCR, which is more appropriate. This is now updated in the article title as well. Please find attached links below for RT - Reverse-Transcriptase, and there are many more available online and this is in common use.
(RT-qPCR Testing of SARS-CoV-2: A Primer - PMC (nih.gov)
(Real-time RT-PCR for COVID-19 diagnosis: challenges and prospects - PMC (nih.gov)
(COVID-19 and Diagnostic Testing for SARS-CoV-2 by RT-qPCR—Facts and Fallacies - PMC (nih.gov)
(Should RT-PCR be considered a gold standard in the diagnosis of COVID-19? - PubMed (nih.gov)
- Line 215 - figure 2 replace with Figure 2 (Actually this is now figure 3).
Response: figure 2 is replaced by Figure 2.
- Chapter 3 - in subsections, replace letters with numbers, for example, 3.1; 3.2. According to the instructions for authors.
Response: Chapter 3- now subsections are changed to 3.1, 3.2 and so on following reviewer suggestion.
Round 2
Reviewer 1 Report
Comments:
Line no. 71 and 72: real-time reverse transcriptase-polymerase chain reaction (RT-qPCR): not correct
Line No. 318: NAAT by RT-qPCR: Some other methods also reported authors should add
Author Response
Reviewer1 (Round2)
We are thankful to the reviewer for their valuable time and appreciate the suggestions, now modified in the text as mentioned below,
- Line no. 71 and 72: real-time reverse transcriptase-polymerase chain reaction (RT-qPCR): not correct
Response: As advised, this is changed to- the reverse transcriptase-polymerase chain reaction (RT-qPCR) technique,
However, this is not incorrect and can be found in the below reference and others on the web,
Xu, J.; Wu, R.; Huang, H.; Zheng, W.; Ren, X.; Wu, N.; Ji, B.; Lv, Y.; Liu, Y.; Mi, R. Computed Tomographic Imaging of 3 Patients With Coronavirus Disease 2019 Pneumonia With Negative Virus Real-time Reverse-Transcription Polymerase Chain Reaction Test. Clin. Infect. Dis. Off. Publ. Infect. Dis. Soc. Am. 2020. [Google Scholar] [CrossRef] [PubMed]
- Line No. 318: NAAT by RT-qPCR: Some other methods also reported authors should add
Response: As advised by the reviewer we have added other NAAT methods in the text as mentioned below,
Recently NAAT has included other techniques such as isothermal amplification platforms with nicking endonuclease amplification reaction (NEAR), loop-mediated isothermal amplification (LAMP), and transcription-mediated amplification (TMA)
Reviewer 2 Report
Thank you very much for all the answers and taking into account most of the comments and making corrections. However, I still cannot agree with line 544 that the method is time consuming and takes 2-5 days. I do not agree with the explanation that transport to the laboratory takes a long time. What does it have to do with it ? You are writing about the method and not about sample transport. One lab will be transporting swabs and the other will have a swab point in the next room. In addition, you can even make one attempt in the reaction, you do not have to wait for a specific amount to accumulate. This is also not true. Please correct this incorrect sentence.
Author Response
Reviewer2 (Round2)
Thank you very much for all the answers and taking into account most of the comments and making corrections. However, I still cannot agree with line 544 that the method is time consuming and takes 2-5 days. I do not agree with the explanation that transport to the laboratory takes a long time. What does it have to do with it ? You are writing about the method and not about sample transport. One lab will be transporting swabs and the other will have a swab point in the next room. In addition, you can even make one attempt in the reaction, you do not have to wait for a specific amount to accumulate. This is also not true. Please correct this incorrect sentence.
We are thankful to the reviewer for their valuable time and appreciate the suggestions, now modified in the text as mentioned below,
Following the reviewer's suggestion, we have now removed the controversial statement. However, the basis for our previous statement can be found in the below reference, which is also cited in the manuscript,
Younes, N.; Al-Sadeq, D.W.; Al-Jighefee, H.; Younes, S.; Al-Jamal, O.; Daas, H.I.; Yassine, H.M.; Nasrallah, G.K. Challenges in Laboratory Diagnosis of the Novel Coronavirus SARS-CoV-2. Viruses 2020, 12, doi:10.3390/v12060582.